# Stochastic Latent Actor-Critic: Deep Reinforcement Learning with a Latent Variable Model

**Alex X. Lee**[1,2]     **Anusha Nagabandi**[1]     **Pieter Abbeel**[1]     **Sergey Levine**[1]

[1]University of California, Berkeley
[2]DeepMind
{alexlee_gk,nagaban2,pabbeel,svlevine}@cs.berkeley.edu

## Abstract

Deep reinforcement learning (RL) algorithms can use high-capacity deep networks to learn directly from image observations. However, these high-dimensional observation spaces present a number of challenges in practice, since the policy must now solve two problems: representation learning and task learning. In this work, we tackle these two problems separately, by explicitly learning latent representations that can accelerate reinforcement learning from images. We propose the stochastic latent actor-critic (SLAC) algorithm: a sample-efficient and high-performing RL algorithm for learning policies for complex continuous control tasks directly from high-dimensional image inputs. SLAC provides a novel and principled approach for unifying stochastic sequential models and RL into a single method, by learning a compact latent representation and then performing RL in the model's learned latent space. Our experimental evaluation demonstrates that our method outperforms both model-free and model-based alternatives in terms of final performance and sample efficiency, on a range of difficult image-based control tasks. Our code and videos of our results are available at our website.[1]

## 1 Introduction

Deep reinforcement learning (RL) algorithms can learn to solve tasks directly from raw, low-level observations such as images. However, such high-dimensional observation spaces present a number of challenges in practice: On one hand, it is difficult to directly learn from these high-dimensional inputs, but on the other hand, it is also difficult to tease out a compact representation of the underlying task-relevant information from which to learn instead. Standard model-free deep RL aims to unify these challenges of representation learning and task learning into a single end-to-end training procedure. However, solving *both* problems together is difficult, since an effective policy requires an effective representation, and an effective representation requires meaningful gradient information to come from the policy or value function, while using only the model-free supervision signal (i.e., the reward function). As a result, learning directly from images with standard end-to-end RL algorithms can in practice be slow, sensitive to hyperparameters, and inefficient.

Instead, we propose to separate representation learning and task learning, by relying on predictive model learning to explicitly acquire a latent representation, and training the RL agent *in that learned latent space*. This alleviates the representation learning challenge because predictive learning benefits from a rich and informative supervision signal even before the agent has made any progress on the task, and thus results in improved sample efficiency of the overall learning process. In this work, our predictive model serves to accelerate task learning by separately addressing representation learning, in contrast to existing model-based RL approaches, which use predictive models either for generating cheap synthetic experience [51, 22, 32] or for planning into the future [11, 13, 46, 9, 55, 26].

Our proposed stochastic sequential model (Figure 1) models the high-dimensional observations as the consequence of a latent process, with a Gaussian prior and latent dynamics. This model represents a partially observed Markov decision process (POMDP), where the stochastic latent state enables the model to represent uncertainty about any of the state variables, given the past observations. Solving such a POMDP exactly would be computationally intractable, since it amounts to solving the decision problem in the space of *beliefs* [5, 33]. Recent works approximate the belief as encodings of latent samples from forward rollouts or particle filtering [8, 30], or as learned belief representations in a belief-state forward model [21]. We instead propose a simple approximation, which we derive from the control as inference framework, that trains a Markovian critic on latent state samples and trains an actor on a history of observations and actions, resulting in our stochastic latent actor-critic (SLAC) algorithm. Although this approximation loses some of the benefits of full POMDP solvers (e.g. reducing uncertainty), it is easy and stable to train in practice, achieving competitive results on a range of challenging problems.

The main contribution of this work is a novel and principled approach that integrates learning stochastic sequential models and RL into a single method, performing RL in the model's learned latent space. By formalizing the problem as a control as inference problem within a POMDP, we show that variational inference leads to the objective of our SLAC algorithm. We empirically show that SLAC benefits from the good asymptotic performance of model-free RL while also leveraging the improved latent space representation for sample efficiency, by demonstrating that SLAC substantially outperforms both prior model-free and model-based RL algorithms on a range of image-based continuous control benchmark tasks.

## 2 Related Work

**Representation learning in RL.** End-to-end deep RL can in principle learn representations implicitly as part of the RL process [45]. However, prior work has observed that RL has a "representation learning bottleneck": a considerable portion of the learning period must be spent acquiring good representations of the observation space [50]. This motivates the use of a distinct representation learning procedure to acquire these representations before the agent has even learned to solve the task. A number of prior works have explored the use of auxiliary supervision in RL to learn such representations [41, 14, 31, 29, 23, 47, 48, 19, 10]. In contrast to this class of representation learning algorithms, we explicitly learn a latent variable model of the POMDP, in which the latent representation and latent-space dynamics are jointly learned. By modeling covariances between consecutive latent states, we make it feasible for our proposed algorithm to perform Bellman backups directly in the latent space of the learned model.

**Partial observability in RL.** Our work is also related to prior research on RL under partial observability. Prior work has studied exact and approximate solutions to POMDPs, but they require explicit models of the POMDP and are only practical for simpler domains [33]. Recent work has proposed end-to-end RL methods that use recurrent neural networks to process histories of observations and (sometimes) actions, but without constructing a model of the POMDP [28, 15, 56]. Other works, however, learn latent-space dynamical system models and then use them to solve the POMDP with model-based RL [54, 53, 34, 35, 55, 26, 36]. Although some of these works learn latent variable models that are similar to ours, these methods are often limited by compounding model errors and finite horizon optimization. In contrast to these works, our approach does not use the model for prediction, and performs infinite horizon policy optimization. Our approach benefits from the good asymptotic performance of model-free RL, while at the same time leveraging the improved latent space representation for sample efficiency.

Other works have also trained latent variable models and used their representations as the inputs to model-free RL algorithms. They use representations encoded from latent states sampled from the forward model [8], belief representations obtained from particle filtering [30], or belief representations obtained directly from a learned belief-space forward model [21]. Our approach is closely related to these prior methods, in that we also use model-free RL with a latent state representation that is learned via prediction. However, instead of using belief representations, our method learns a critic directly on latent state samples, which more tractably enables scaling to more complex tasks. Concurrent to our work, Hafner et al. [27] proposed to integrate model-free learning with representations from sequence models, as proposed in this paper, with model-based rollouts, further improving on the performance of prior model-based approaches.

**Sequential latent variable models.** Several previous works have explored various modeling choices to learn stochastic sequential models [40, 4, 34, 16, 17, 12, 20]. They vary in the factorization of the generative and inference models, their network architectures, and the objectives used in their training procedures. Our approach is compatible with any of these sequential latent variable models, with the only requirement being that they provide a mechanism to sample latent states from the belief of the learned Markovian latent space.

## 3 Preliminaries

This work addresses the problem of learning policies from high-dimensional observations in POMDPs, by simultaneously learning a latent representation of the underlying MDP state using variational inference, as well as learning a policy in a maximum entropy RL framework. In this section, we describe maximum entropy RL [57, 24, 42] in fully observable MDPs, as well as variational methods for training latent state space models for POMDPs.

### 3.1 Maximum Entropy RL in Fully Observable MDPs

Consider a Markov decision process (MDP), with states $\mathbf{s}_t \in \mathcal{S}$, actions $\mathbf{a}_t \in \mathcal{A}$, rewards $r_t$, initial state distribution $p(\mathbf{s}_1)$, and stochastic transition distribution $p(\mathbf{s}_{t+1}|\mathbf{s}_t, \mathbf{a}_t)$. Standard RL aims to learn the parameters $\phi$ of some policy $\pi_\phi(\mathbf{a}_t|\mathbf{s}_t)$ such that the expected sum of rewards is maximized under the induced trajectory distribution $\rho_\pi$. This objective can be modified to incorporate an *entropy* term, such that the policy also aims to maximize the expected entropy $\mathcal{H}(\pi_\phi(\cdot|\mathbf{s}_t))$. This formulation has a close connection to variational inference [57, 24, 42], and we build on this in our work. The resulting maximum entropy objective is $\sum_{t=1}^{T} \mathbb{E}_{(\mathbf{s}_t, \mathbf{a}_t) \sim \rho_\pi} [r(\mathbf{s}_t, \mathbf{a}_t) + \alpha \mathcal{H}(\pi_\phi(\cdot|\mathbf{s}_t))]$, where $r$ is the reward function, and $\alpha$ is a temperature parameter that trades off between maximizing for the reward and for the policy entropy. Soft actor-critic (SAC) [24] uses this maximum entropy RL framework to derive soft policy iteration, which alternates between policy evaluation and policy improvement within the described maximum entropy framework. SAC then extends this soft policy iteration to handle continuous action spaces by using parameterized function approximators to represent both the Q-function $Q_\theta$ (critic) and the policy $\pi_\phi$ (actor). The soft Q-function parameters $\theta$ are optimized to minimize the soft Bellman residual,

$$J_Q(\theta) = \tfrac{1}{2} \left( Q_\theta(\mathbf{s}_t, \mathbf{a}_t) - \left( r_t + \gamma \mathop{\mathbb{E}}_{\mathbf{a}_{t+1} \sim \pi_\phi} [Q_{\bar{\theta}}(\mathbf{s}_{t+1}, \mathbf{a}_{t+1}) - \alpha \log \pi_\phi(\mathbf{a}_{t+1}|\mathbf{s}_{t+1})] \right) \right)^2, \quad (1)$$

where $\gamma$ is the discount factor, and $\bar{\theta}$ are delayed parameters. The policy parameters $\phi$ are optimized to update the policy towards the exponential of the soft Q-function, resulting in the policy loss

$$J_\pi(\phi) = \mathop{\mathbb{E}}_{\mathbf{a}_t \sim \pi_\phi} [\alpha \log(\pi_\phi(\mathbf{a}_t|\mathbf{s}_t)) - Q_\theta(\mathbf{s}_t, \mathbf{a}_t)]. \quad (2)$$

SLAC builds on top of this maximum entropy RL framework, by further integrating explicit representation learning and handling partial observability.

### 3.2 Sequential Latent Variable Models and Amortized Variational Inference in POMDPs

To learn representations for RL, we use latent variable models trained with amortized variational inference. The learned model must be able to process a large number of pixels that are present in the entangled image $\mathbf{x}$, and it must tease out the relevant information into a compact and disentangled representation $\mathbf{z}$. To learn such a model, we can consider maximizing the probability of each observed datapoint $\mathbf{x}$ from some training set under the entire generative process $p(\mathbf{x}) = \int p(\mathbf{x}|\mathbf{z})p(\mathbf{z}) \, d\mathbf{z}$. This objective is intractable to compute in general due to the marginalization of the latent variables $\mathbf{z}$. In amortized variational inference, we utilize the evidence lower bound for the log-likelihood [38]:

$$\log p(\mathbf{x}) \geq E_{\mathbf{z} \sim q} [\log p(\mathbf{x}|\mathbf{z})] - \mathrm{D}_{\mathrm{KL}}(q(\mathbf{z}|\mathbf{x}) \,\|\, p(\mathbf{z})). \quad (3)$$

We can maximize the probability of the observed datapoints (i.e., the left hand side of Equation (3)) by learning an encoder $q(\mathbf{z}|\mathbf{x})$ and a decoder $p(\mathbf{x}|\mathbf{z})$, and then directly performing gradient ascent on the right hand side of the equation. In this setup, the distributions of interest are the prior $p(\mathbf{z})$, the observation model $p(\mathbf{x}|\mathbf{z})$, and the variational approximate posterior $q(\mathbf{z}|\mathbf{x})$.

In order to extend such models to sequential decision making settings, we must incorporate actions and impose temporal structure on the latent state. Consider a partially observable MDP (POMDP), with latent states $\mathbf{z}_t \in \mathcal{Z}$ and its corresponding observations $\mathbf{x}_t \in \mathcal{X}$. We make an explicit distinction between an observation $\mathbf{x}_t$ and the underlying latent state $\mathbf{z}_t$, to emphasize that the latter is unobserved and its distribution is unknown. Analogous to the MDP, the initial and transition distributions are $p(\mathbf{z}_1)$ and $p(\mathbf{z}_{t+1}|\mathbf{z}_t, \mathbf{a}_t)$, and the reward is $r_t$. In addition, the observation model is given by $p(\mathbf{x}_t|\mathbf{z}_t)$.

As in the case for VAEs, a generative model of these observations $\mathbf{x}_t$ can be learned by maximizing the log-likelihood. In the POMDP setting, however, we note that $\mathbf{x}_t$ alone does not provide all necessary information to infer $\mathbf{z}_t$, and prior observations must be taken into account during inference. This brings us to the discussion of sequential latent variable models. The distributions of interest are $p(\mathbf{z}_1)$ and $p(\mathbf{z}_{t+1}|\mathbf{z}_t, \mathbf{a}_t)$, the observation model $p(\mathbf{x}_t|\mathbf{z}_t)$, and the approximate variational posteriors $q(\mathbf{z}_1|\mathbf{x}_1)$ and $q(\mathbf{z}_{t+1}|\mathbf{x}_{t+1}, \mathbf{z}_t, \mathbf{a}_t)$. The log-likelihood of the observations can then be bounded,

$$\log p(\mathbf{x}_{1:\tau+1}|\mathbf{a}_{1:\tau}) \geq \mathop{\mathbb{E}}_{\mathbf{z}_{1:\tau+1}\sim q}\left[\sum_{t=0}^{\tau}\log p(\mathbf{x}_{t+1}|\mathbf{z}_{t+1}) - D_{\mathrm{KL}}(q(\mathbf{z}_{t+1}|\mathbf{x}_{t+1}, \mathbf{z}_t, \mathbf{a}_t) \parallel p(\mathbf{z}_{t+1}|\mathbf{z}_t, \mathbf{a}_t))\right]. \tag{4}$$

For notational convenience, we define $q(\mathbf{z}_1|\mathbf{x}_1, \mathbf{z}_0, \mathbf{a}_0) \coloneqq q(\mathbf{z}_1|\mathbf{x}_1)$ and $p(\mathbf{z}_1|\mathbf{z}_0, \mathbf{a}_0) \coloneqq p(\mathbf{z}_1)$. Prior work [8, 30, 21, 26, 20, 36, 12, 55] has explored modeling such non-Markovian observation sequences, using methods such as recurrent neural networks with deterministic hidden state, as well as probabilistic state-space models. In this work, we enable the effective training of a fully stochastic sequential latent variable model, and bring it together with a maximum entropy actor-critic RL algorithm to create SLAC: a sample-efficient and high-performing RL algorithm for learning policies for complex continuous control tasks directly from high-dimensional image inputs.

## 4   Joint Modeling and Control as Inference

For a fully observable MDP, the control problem can be embedded into a graphical model by introducing a binary random variable $\mathcal{O}_t$, which indicates if time step $t$ is optimal. When its distribution is chosen to be $p(\mathcal{O}_t = 1|\mathbf{s}_t, \mathbf{a}_t) = \exp(r(\mathbf{s}_t, \mathbf{a}_t))$, then maximization of $p(\mathcal{O}_{1:T})$ via approximate inference in that model yields the optimal policy for the maximum entropy objective [42].

In this paper, we extend this idea to the POMDP setting, where the probabilistic graphical model includes latent variables, as shown in Figure 1, and the distribution can analogously be given by $p(\mathcal{O}_t = 1|\mathbf{z}_t, \mathbf{a}_t) = \exp(r(\mathbf{z}_t, \mathbf{a}_t))$. Instead of maximizing the likelihood of the optimality variables alone, we jointly model the observations (including the observed rewards of the past time steps) and learn maximum entropy policies by maximizing the marginal likelihood $p(\mathbf{x}_{1:\tau+1}, \mathcal{O}_{\tau+1:T}|\mathbf{a}_{1:\tau})$. This objective represents both the likelihood of the observed data from the past $\tau + 1$ steps, as well as the optimality of the agent's actions for future steps, effectively combining both

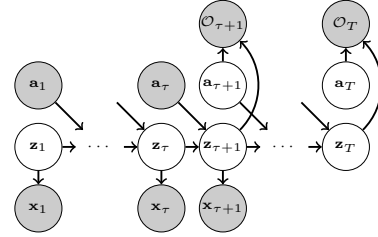

**Figure 1:** Graphical model of POMDP with optimality variables for $t \geq \tau + 1$.

representation learning and control into a single graphical model. We factorize our variational distribution into a product of *recognition* terms $q(\mathbf{z}_{t+1}|\mathbf{x}_{t+1}, \mathbf{z}_t, \mathbf{a}_t)$, *dynamics* terms $p(\mathbf{z}_{t+1}|\mathbf{z}_t, \mathbf{a}_t)$, and *policy* terms $\pi(\mathbf{a}_t|\mathbf{x}_{1:t}, \mathbf{a}_{1:t-1})$:

$$q(\mathbf{z}_{1:T}, \mathbf{a}_{\tau+1:T}|\mathbf{x}_{1:\tau+1}, \mathbf{a}_{1:\tau}) = \prod_{t=0}^{\tau} q(\mathbf{z}_{t+1}|\mathbf{x}_{t+1}, \mathbf{z}_t, \mathbf{a}_t) \prod_{t=\tau+1}^{T-1} p(\mathbf{z}_{t+1}|\mathbf{z}_t, \mathbf{a}_t) \prod_{t=\tau+1}^{T} \pi(\mathbf{a}_t|\mathbf{x}_{1:t}, \mathbf{a}_{1:t-1}). \tag{5}$$

The variational distribution uses the dynamics for future time steps to prevent the agent from controlling the transitions and from choosing optimistic actions, analogously to the fully observed MDP setting described by Levine [42]. The posterior over the actions represents the policy $\pi$.

We use the posterior from Equation (5) to obtain the evidence lower bound (ELBO) of the likelihood,

$$\log p(\mathbf{x}_{1:\tau+1}, \mathcal{O}_{\tau+1:T}|\mathbf{a}_{1:\tau})$$

$$\geq \mathop{\mathbb{E}}_{(\mathbf{z}_{1:T}, \mathbf{a}_{\tau+1:T})\sim q}\left[\log p(\mathbf{x}_{1:\tau+1}, \mathcal{O}_{\tau+1:T}, \mathbf{z}_{1:T}, \mathbf{a}_{\tau+1:T}|\mathbf{a}_{1:\tau}) - \log q(\mathbf{z}_{1:T}, \mathbf{a}_{\tau+1:T}|\mathbf{x}_{1:\tau+1}, \mathbf{a}_{1:\tau})\right]$$

$$= \mathop{\mathbb{E}}_{(\mathbf{z}_{1:T}, \mathbf{a}_{\tau+1:T})\sim q}\left[\underbrace{\sum_{t=0}^{\tau}\Big(\log p(\mathbf{x}_{t+1}|\mathbf{z}_{t+1}) - D_{\mathrm{KL}}(q(\mathbf{z}_{t+1}|\mathbf{x}_{t+1}, \mathbf{z}_t, \mathbf{a}_t) \parallel p(\mathbf{z}_{t+1}|\mathbf{z}_t, \mathbf{a}_t))\Big)}_{\text{model objective terms}} \right.$$

$$\left. + \underbrace{\sum_{t=\tau+1}^{T}\Big(r(\mathbf{z}_t, \mathbf{a}_t) + \log p(\mathbf{a}_t) - \log \pi(\mathbf{a}_t|\mathbf{x}_{1:t}, \mathbf{a}_{1:t-1})\Big)}_{\text{policy objective terms}}\right], \tag{6}$$

where $r(\mathbf{z}_t, \mathbf{a}_t) = \log p(\mathcal{O}_t = 1|\mathbf{z}_t, \mathbf{a}_t)$ by construction and $p(\mathbf{a}_t)$ is the action prior. The full derivation of the ELBO is given in Appendix A.

# 5 Stochastic Latent Actor Critic

We now describe our stochastic latent actor critic (SLAC) algorithm, which maximizes the ELBO using function approximators to model the prior and posterior distributions. The ELBO objective in Equation (6) can be split into a model objective and a maximum entropy RL objective. The model objective can be optimized directly, while the maximum entropy RL objective can be optimized via approximate message passing, with messages corresponding to the Q-function. We can rewrite the RL objective to express it in terms of these messages, yielding an actor-critic algorithm analogous to SAC. Additional details of the derivation of the SLAC objectives are given in Appendix A.

**Latent variable model.** The first part of the ELBO corresponds to training the latent variable model to maximize the likelihood of the observations, analogous to the ELBO in Equation (4) for the sequential latent variable model. The generative model is given by $p_\psi(\mathbf{z}_1)$, $p_\psi(\mathbf{z}_{t+1}|\mathbf{z}_t, \mathbf{a}_t)$, and $p_\psi(\mathbf{x}_t|\mathbf{z}_t)$, and the inference model is given by $q_\psi(\mathbf{z}_1|\mathbf{x}_1)$ and $q_\psi(\mathbf{z}_{t+1}|\mathbf{x}_{t+1}, \mathbf{z}_t, \mathbf{a}_t)$. These distributions are diagonal Gaussian, where the means and variances are given by outputs of neural networks. Further details of our specific model architecture are given in Appendix B. The distribution parameters $\psi$ are optimized with respect to the ELBO in Equation (6), where the only terms that depend on $\psi$, and therefore constitute the model objective, are given by

$$J_M(\psi) = \mathop{\mathbb{E}}_{\mathbf{z}_{1:\tau+1} \sim q_\psi} \left[ \sum_{t=0}^{\tau} - \log p_\psi(\mathbf{x}_{t+1}|\mathbf{z}_{t+1}) + D_{\text{KL}}(q_\psi(\mathbf{z}_{t+1}|\mathbf{x}_{t+1}, \mathbf{z}_t, \mathbf{a}_t) \| p_\psi(\mathbf{z}_{t+1}|\mathbf{z}_t, \mathbf{a}_t)) \right], \quad (7)$$

where we define $q_\psi(\mathbf{z}_1|\mathbf{x}_1, \mathbf{z}_0, \mathbf{a}_0) := q_\psi(\mathbf{z}_1|\mathbf{x}_1)$ and $p_\psi(\mathbf{z}_1|\mathbf{z}_0, \mathbf{a}_0) := p_\psi(\mathbf{z}_1)$. We use the reparameterization trick to sample from the filtering distribution $q_\psi(\mathbf{z}_{1:\tau+1}|\mathbf{x}_{1:\tau+1}, \mathbf{a}_{1:\tau})$.

**Actor and critic.** The second part of the ELBO corresponds to the maximum entropy RL objective. As in the fully observable case from Section 3.1 and as described by Levine [42], this optimization can be solved via message passing of soft Q-values. However, in our method, we must use the latent states $\mathbf{z}$, since the true state is unknown. The messages are approximated by minimizing the soft Bellman residual, which we use to train our soft Q-function parameters $\theta$,

$$J_Q(\theta) = \mathop{\mathbb{E}}_{\mathbf{z}_{1:\tau+1} \sim q_\psi} \left[ \tfrac{1}{2} \left( Q_\theta(\mathbf{z}_\tau, \mathbf{a}_\tau) - (r_\tau + \gamma V_{\bar{\theta}}(\mathbf{z}_{\tau+1})) \right)^2 \right], \quad (8)$$

$$V_\theta(\mathbf{z}_{\tau+1}) = \mathop{\mathbb{E}}_{\mathbf{a}_{\tau+1} \sim \pi_\phi} \left[ Q_\theta(\mathbf{z}_{\tau+1}, \mathbf{a}_{\tau+1}) - \alpha \log \pi_\phi(\mathbf{a}_{\tau+1}|\mathbf{x}_{1:\tau+1}, \mathbf{a}_{1:\tau}) \right], \quad (9)$$

where $V_\theta$ is the soft state value function and $\bar{\theta}$ are delayed target network parameters, obtained as exponential moving averages of $\theta$. Notice that the latents $\mathbf{z}_\tau$ and $\mathbf{z}_{\tau+1}$, which are used in the Bellman backup, are sampled from the same filtering distribution, i.e. $\mathbf{z}_{\tau+1} \sim q_\psi(\mathbf{z}_{\tau+1}|\mathbf{x}_{\tau+1}, \mathbf{z}_\tau, \mathbf{a}_\tau)$. The RL objective, which corresponds to the second part of the ELBO, can then be rewritten in terms of the soft Q-function. The policy parameters $\phi$ are optimized to maximize this objective, resulting in a policy loss analogous to soft actor-critic [24]:

$$J_\pi(\phi) = \mathop{\mathbb{E}}_{\mathbf{z}_{1:\tau+1} \sim q_\psi} \left[ \mathop{\mathbb{E}}_{\mathbf{a}_{\tau+1} \sim \pi_\phi} \left[ \alpha \log \pi_\phi(\mathbf{a}_{\tau+1}|\mathbf{x}_{1:\tau+1}, \mathbf{a}_{1:\tau}) - Q_\theta(\mathbf{z}_{\tau+1}, \mathbf{a}_{\tau+1}) \right] \right]. \quad (10)$$

We assume a uniform action prior, so $\log p(\mathbf{a}_t)$ is a constant term that we omit from the policy loss. This loss only uses the last sample $\mathbf{z}_{\tau+1}$ of the sequence for the critic, and we use the reparameterization trick to sample from the policy. Note that the policy is not conditioned on the latent state, as this can lead to over-optimistic behavior since the algorithm would learn Q-values for policies that have perfect access to the latent state. Instead, the learned policy in our algorithm is conditioned directly on the past observations and actions. This has the additional benefit that the learned policy can be executed at run time without requiring inference of the latent state. Finally, we note that for the expectation over latent states in the Bellman residual in Equation (9), rather than sampling latent states for all $\mathbf{z} \sim \mathcal{Z}$, we sample latent states from the filtering distribution $q_\psi(\mathbf{z}_{1:\tau+1}|\mathbf{x}_{1:\tau+1}, \mathbf{a}_{1:\tau})$. This design choice allows us to minimize the critic loss for samples that are most relevant for $Q_\theta$, while also allowing the critic loss to use the Q-function in the same way as implied by the policy loss in Equation (10).

---

**Algorithm 1** Stochastic Latent Actor-Critic (SLAC)

---

**Require:** Environment $E$ and initial parameters $\psi, \phi, \theta_1, \theta_2$ for the model, actor, and critics.
  $\mathbf{x}_1 \sim E_{\text{reset}}()$
  $\mathcal{D} \leftarrow (\mathbf{x}_1)$
  **for** each iteration **do**
    **for** each environment step **do**
      $\mathbf{a}_t \sim \pi_\phi(\mathbf{a}_t|\mathbf{x}_{1:t}, \mathbf{a}_{1:t-1})$
      $r_t, \mathbf{x}_{t+1} \sim E_{\text{step}}(\mathbf{a}_t)$
      $\mathcal{D} \leftarrow \mathcal{D} \cup (\mathbf{a}_t, r_t, \mathbf{x}_{t+1})$
    **for** each gradient step **do**
      $\mathbf{x}_{1:\tau+1}, \mathbf{a}_{1:\tau}, r_\tau \sim \mathcal{D}$
      $\mathbf{z}_{1:\tau+1} \sim q_\psi(\mathbf{z}_{1:\tau+1}|\mathbf{x}_{1:\tau+1}, \mathbf{a}_{1:\tau})$
      $\psi \leftarrow \psi - \lambda_M \nabla_\psi J_M(\psi)$
      $\theta_i \leftarrow \theta_i - \lambda_Q \nabla_{\theta_i} J_Q(\theta_i)$ for $i \in \{1, 2\}$
      $\phi \leftarrow \phi - \lambda_\pi \nabla_\phi J_\pi(\phi)$
      $\bar{\theta}_i \leftarrow \nu \theta_i + (1 - \nu) \bar{\theta}_i$ for $i \in \{1, 2\}$

---

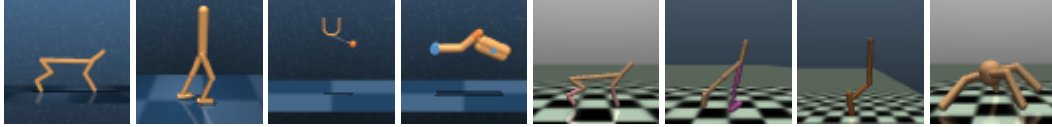

**Figure 2:** Example image observations for our continuous control benchmark tasks: DeepMind Control's cheetah run, walker walk, ball-in-cup catch, and finger spin, and OpenAI Gym's half cheetah, walker, hopper, and ant (left to right). These images, which are rendered at a resolution of $64 \times 64$ pixels, are the observation inputs to our algorithm, i.e. to the latent variable model and to the policy.

SLAC is outlined in Algorithm 1. The actor-critic component follows prior work, with automatic tuning of the temperature $\alpha$ and two Q-functions to mitigate overestimation [18, 24, 25]. SLAC can be viewed as a variant of SAC [24] where the critic is trained on the stochastic latent state of our sequential latent variable model. The backup for the critic is performed on a tuple $(\mathbf{z}_\tau, \mathbf{a}_\tau, r_\tau, \mathbf{z}_{\tau+1})$, sampled from the filtering distribution $q_\psi(\mathbf{z}_{\tau+1}, \mathbf{z}_\tau | \mathbf{x}_{1:\tau+1}, \mathbf{a}_{1:\tau})$. The critic can, in principle, take advantage of the perfect knowledge of the state $\mathbf{z}_t$, which makes learning easier. However, the policy does not have access to $\mathbf{z}_t$, and must make decisions based on a history of observations and actions. SLAC is not a model-based algorithm, in that in does not use the model for prediction, but we see in our experiments that SLAC can achieve similar sample efficiency as a model-based algorithm.

## 6   Experimental Evaluation

We evaluate SLAC on multiple image-based continuous control tasks from both the DeepMind Control Suite [52] and OpenAI Gym [7], as illustrated in Figure 2. Full details of SLAC's network architecture are described in Appendix B. Training and evaluation details are given in Appendix C, and image samples from our model for all tasks are shown in Appendix E. Additionally, visualizations of our results and code are available on the project website.[2]

### 6.1   Comparative Evaluation on Continuous Control Benchmark Tasks

To provide a comparative evaluation against prior methods, we evaluate SLAC on four tasks (cheetah run, walker walk, ball-in-cup catch, finger spin) from the DeepMind Control Suite [52], and four tasks (cheetah, walker, ant, hopper) from OpenAI Gym [7]. Note that the Gym tasks are typically used with low-dimensional state observations, while we evaluate on them with raw image observations. We compare our method to the following state-of-the-art model-based and model-free algorithms:

**SAC** [24]: This is an off-policy actor-critic algorithm, which represents a comparison to state-of-the-art model-free learning. We include experiments showing the performance of SAC based on true state (as an upper bound on performance) as well as directly from raw images.

**D4PG** [6]: This is also an off-policy actor-critic algorithm, learning directly from raw images. The results reported in the plots below are the performance after $10^8$ training steps, as stated in the benchmarks from Tassa et al. [52].

**MPO** [2, 1]: This is an off-policy actor-critic algorithm that performs an expectation maximization form of policy iteration, learning directly from raw images.

**DVRL** [30]: This is an on-policy model-free RL algorithm that trains a partially stochastic latent-variable POMDP model. DVRL uses the *full belief* over the latent state as input into both the actor and critic, as opposed to our method, which trains the critic with the latent state and the actor with a history of actions and observations.

**PlaNet** [26]: This is a model-based RL method for learning from images, which uses a partially stochastic sequential latent variable model, but without explicit policy learning. Instead, the model is used for planning with model predictive control (MPC), where each plan is optimized with the cross entropy method (CEM).

**DrQ** [39]: This is the same as the SAC algorithm, but combined with data augmentation on the image inputs.

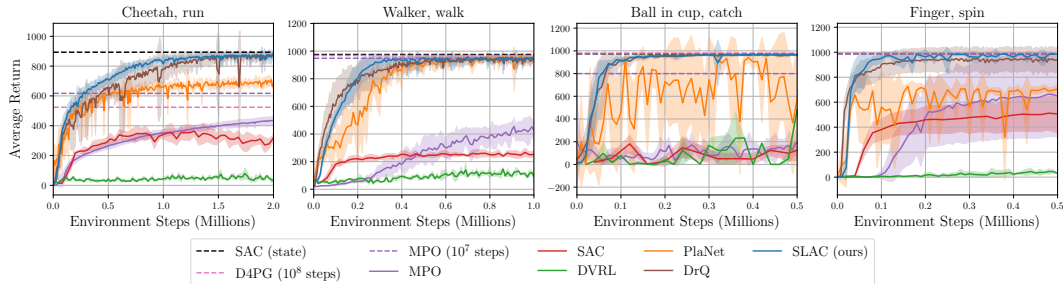

**Figure 3:** Experiments on the DeepMind Control Suite from images (unless otherwise labeled as "state"). SLAC (ours) converges to similar or better final performance than the other methods, while almost always achieving reward as high as the upper bound SAC baseline that learns from true state. Note that for these experiments, 1000 environments steps corresponds to 1 episode.

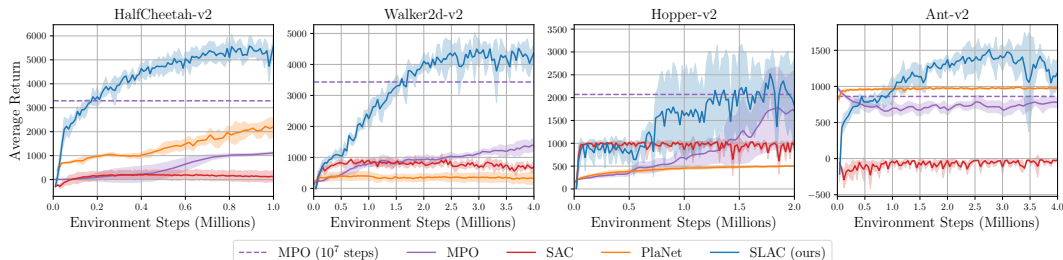

**Figure 4:** Experiments on the OpenAI Gym benchmark tasks from images. SLAC (ours) converges to higher performance than both PlaNet and SAC on all four of these tasks. The number of environments steps in each episode is variable, depending on the termination.

Our experiments on the DeepMind Control Suite in Figure 3 show that the sample efficiency of SLAC is comparable or better than *both* model-based and model-free alternatives. This indicates that overcoming the representation learning bottleneck, coupled with efficient off-policy RL, provides for fast learning similar to model-based methods, while attaining final performance comparable to fully model-free techniques that learn from state. SLAC also substantially outperforms DVRL. This difference can be explained in part by the use of an efficient off-policy RL algorithm, which can better take advantage of the learned representation. SLAC achieves comparable or slightly better performance than subsequent work DrQ, which also uses the efficient off-policy SAC algorithm.

We also evaluate SLAC on continuous control benchmark tasks from OpenAI Gym in Figure 4. We notice that these tasks are more challenging than the DeepMind Control Suite tasks, because the rewards are not as shaped and not bounded between 0 and 1, the dynamics are different, and the episodes terminate on failure (e.g., when the hopper or walker falls over). PlaNet is unable to solve the last three tasks, while for the cheetah task, it learns a suboptimal policy that involves flipping the cheetah over and pushing forward while on its back. To better understand the performance of fixed-horizon MPC on these tasks, we also evaluated with the ground truth dynamics (i.e., the true simulator), and found that even in this case, MPC did not achieve good final performance, suggesting that infinite horizon policy optimization, of the sort performed by SLAC and model-free algorithms, is important to attain good results on these tasks.

Our experiments show that SLAC successfully learns complex continuous control benchmark tasks from raw image inputs. On the DeepMind Control Suite, SLAC exceeds the performance of prior work PlaNet on the four tasks, and SLAC achieves comparable or slightly better performance than subsequence work DrQ. However, on the harder image-based OpenAI Gym tasks, SLAC outperforms PlaNet by a large margin. We note that the prior methods that we tested generally performed poorly on the image-based OpenAI Gym tasks, despite considerable hyperparameter tuning.

## 6.2 Ablation Experiments

We investigate how SLAC is affected by the choice of latent variable model, the inputs given to the actor and critic, the model pretraining, and the number of training updates relative to the number of agent interactions. Additional results are given in Appendix D, including experiments that compare the effect of the decoder output variance and using random cropping for data augmentation.

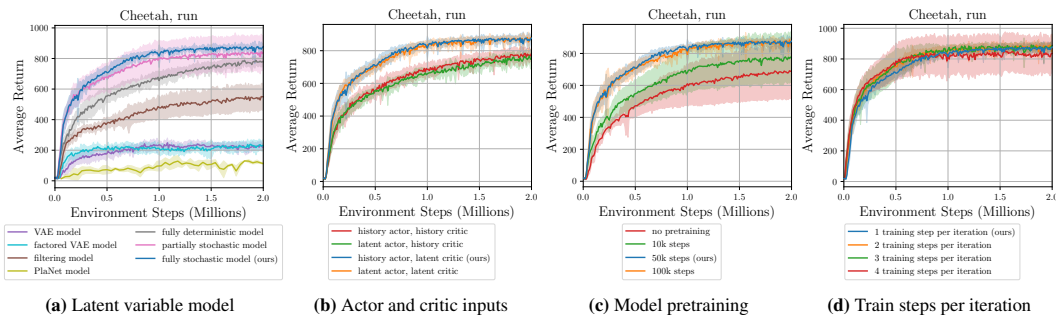

**Figure 5:** Comparison of different design choices for (a) the latent variable model, (b) the inputs given to the actor and critic, either the history of past observations and actions, or a latent sample, (c) the number of model pretraining steps, and (d) the number of training updates per iteration. In all cases, we use the RL framework of SLAC. See Figure 8, Figure 9, Figure 10, and Figure 11 for results on 5 additional tasks.

**Latent variable model.** We study the tradeoffs between different design choices for the latent variable model in Figure 5a and Figure 8. We compare our *fully stochastic* model to a standard non-sequential *VAE* model [38], which has been used in multiple prior works for representation learning in RL [29, 23, 47], and a non-sequential *factored VAE* model, which uses our autoregressive two-variable factorization but without any temporal dependencies. We also compare to a sequential *filtering* model that uses temporal dependencies but without the two-variable factorization, the partially stochastic model used by *PlaNet* [26], as well as two additional variants of our model: a *fully deterministic* model that removes all stochasticity from the hidden state dynamics, and a *partially stochastic* model that adds deterministic paths in the transitions, similar to the PlaNet model, but with our latent factorization and architecture. All the models, except for the PlaNet model, are variants of our model that use the same architecture as our fully stochastic model, with minimal differences in the transitions or the latent variable factorization. In all cases, we use the RL framework of SLAC and only vary the choice of model for representation learning.

Our fully stochastic model outperforms all the other models. Contrary to the conclusions in prior work [26, 8], the fully stochastic model slightly outperforms the partially stochastic model, while retaining the appealing interpretation of a stochastic state space model. We hypothesize that these prior works benefit from the deterministic paths (realized as an LSTM or GRU) because they use multi-step samples from the prior. In contrast, our method uses samples from the posterior, which are conditioned on same-step observations, and thus it is less sensitive to the propagation of the latent states through time. The sequential variants of our model (including ours) outperform the non-sequential VAE models. The models with the two-variable factorization perform similarly or better than their respective equivalents among the non-sequential VAE models and among the sequential stochastic models. Overall, including temporal dependencies results in the largest improvement in performance, followed by the autoregressive latent variable factorization and using a fully stochastic model.

**Actor and critic inputs.** We next investigate alternative choices for the actor and critic inputs as either the observation-action history or the latent sample. In SLAC, the actor is conditioned on the observation-action history and the critic is conditioned on individual latent samples. The images in the history are first compressed with the model's convolutional network before they are given to the networks. However, the actor and critic losses do not propagate any gradient signal into the model nor its convolutional layers, i.e. the convolutional layers used for the observation-action history are only trained by the model loss.

Figure 5b and Figure 9 show that, in general, the performance is significantly worse when the critic input is the history instead of the latent sample, and indifferent to the choice for the actor input. This is consistent with our derivation—the critic should be given latent samples, but the actor can be conditioned on anything (since the policy is the variational posterior). However, we note that a latent-conditioned actor could lead to overconfident behaviors in uncertain environments. For generality, we choose to give the raw history directly to the actor.

**Model pretraining.** We next study the effect of pretraining the model before the agent starts learning on the task. In our experiments, the agent first collects a small amount of data by executing random actions, and then the model is pretrained with that data. The model is pretrained for 50000 iterations on the DeepMind Control Suite experiments, unless otherwise specified. Figure 5c and Figure 10

show that little or no pretraining results in slower learning and, in some cases, worse asymptotic performance. There is almost no difference in performance when using 100000 instead of 50000 iterations, although the former resulted in higher variance across trials in some of the tasks. Overall, these results show that the agent benefits from the supervision signal of the model even before the agent has made any progress on the task.

**Training updates per iteration.** We next investigate the effect of the number of training updates per iteration, or equivalently, the number of training updates per environment step (we use 1 environment step per iteration in all of our experiments). Figure 5d and Figure 11 show that, in general, more training updates per iteration speeds up learning slightly, but too many updates per iteration causes higher variance across trials and slightly worse asymptotic performance in some tasks. Nevertheless, this drop in asymptotic performance (if any) is small, which indicates that our method is less susceptible to overfitting compared to methods in prior work. We hypothesize that using stochastic latent samples to train the critic provides some randomization, which limits overfitting. The best tradeoff is achieved when using 2 training updates per iteration, however, in line with other works, we use 1 training update per iteration in all the other experiments.

# 7   Conclusion

We presented SLAC, an efficient RL algorithm for learning from high-dimensional image inputs that combines efficient off-policy model-free RL with representation learning via a sequential stochastic state space model. Through representation learning in conjunction with effective task learning in the learned latent space, our method achieves improved sample efficiency and final task performance as compared to both prior model-based and model-free RL methods.

While our current SLAC algorithm is fully model-free, in that predictions from the model are not utilized to speed up training, a natural extension of our approach would be to use the model predictions themselves to generate synthetic samples. Incorporating this additional synthetic model-based data into a mixed model-based and model-free method could further improve sample efficiency and performance. More broadly, the use of explicit representation learning with RL has the potential to not only accelerate training time and increase the complexity of achievable tasks, but also enable reuse and transfer of our learned representation across tasks.

# Broader Impact

Despite the existence of automated robotic systems in controlled environments such as factories or labs, standard approaches to controlling systems still require precise and expensive sensor setups to monitor the relevant details of interest in the environment, such as the joint positions of a robot or pose information of all objects in the area. To instead be able to learn directly from the more ubiquitous and rich modality of vision would greatly advance the current state of our learning systems. Not only would this ability to learn directly from images preclude expensive real-world setups, but it would also remove the expensive need for human-engineering efforts in state estimation. While it would indeed be very beneficial for our learning systems to be able to learn directly from raw image observations, this introduces algorithm challenges of dealing with high-dimensional as well as partially observable inputs. In this paper, we study the use of explicitly learning latent representations to assist model-free reinforcement learning directly from raw, high-dimensional images.

Standard end-to-end RL methods try to solve both representation learning and task learning together, and in practice, this leads to brittle solutions which are sensitive to hyperparameters but are also slow and inefficient. These challenges illustrate the predominant use of simulation in the deep RL community; we hope that with more efficient, stable, easy-to-use, and easy-to-train deep RL algorithms such as the one we propose in this work, we can help the field of deep RL to transition to more widespread use in real-world setups such as robotics.

From a broader perspective, there are numerous use cases and areas of application where autonomous decision making agents can have positive effects in our society, from automating dangerous and undesirable tasks, to accelerating automation and economic efficiency of society. That being said, however, automated decision making systems do introduce safety concerns, further exacerbated by the lack of explainability when they do make mistakes. Although this work does not explicitly address safety concerns, we feel that it can be used in conjunction with levels of safety controllers to minimize negative impacts, while drawing on its powerful deep reinforcement learning roots to enable automated and robust tasks in the real world.

## Acknowledgments and Disclosure of Funding

We thank Marvin Zhang, Abhishek Gupta, and Chelsea Finn for useful discussions and feedback, Danijar Hafner for providing timely assistance with PlaNet, and Maximilian Igl for providing timely assistance with DVRL. This research was supported by the National Science Foundation through IIS-1651843 and IIS-1700697, as well as ARL DCIST CRA W911NF-17-2-0181 and the Office of Naval Research. Compute support was provided by NVIDIA.

## Footnotes

[1]https://alexlee-gk.github.io/slac/

[2]`https://alexlee-gk.github.io/slac/`

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
