[Supplementary Material]

# A  Derivation of the Evidence Lower Bound and SLAC Objectives

In this appendix, we discuss how the SLAC objectives can be derived from applying a variational inference scheme to the control as inference framework for reinforcement learning [42]. In this framework, the problem of finding the optimal policy is cast as an inference problem, conditioned on the evidence that the agent is behaving optimally. While Levine [42] derives this in the fully observed case, we present a derivation in the POMDP setting. For reference, we reproduce the probabilistic graphical model in Figure 6.

**Figure 6:** Graphical model of POMDP with optimality variables for $t \geq \tau + 1$.

We aim to maximize the marginal likelihood $p(\mathbf{x}_{1:\tau+1}, \mathcal{O}_{\tau+1:T} | \mathbf{a}_{1:\tau})$, where $\tau$ is the number of steps that the agent has already taken. This likelihood reflects that the agent cannot modify the past $\tau$ actions and they might have not been optimal, but it can choose the future actions up to the end of the episode, such that the chosen future actions are optimal. Notice that unlike the standard control as inference framework, in this work we not only maximize the likelihood of the optimality variables but also the likelihood of the observations, which provides additional supervision for the latent representation. This does not come up in the MDP setting since the state representation is fixed and learning a dynamics model of the state would not change the model-free equations derived from the maximum entropy RL objective.

For reference, we restate the factorization of our variational distribution:

$$
q(\mathbf{z}_{1:T}, \mathbf{a}_{\tau+1:T} | \mathbf{x}_{1:\tau+1}, \mathbf{a}_{1:\tau}) = \prod_{t=0}^{\tau} q(\mathbf{z}_{t+1} | \mathbf{x}_{t+1}, \mathbf{z}_t, \mathbf{a}_t) \prod_{t=\tau+1}^{T-1} p(\mathbf{z}_{t+1} | \mathbf{z}_t, \mathbf{a}_t) \prod_{t=\tau+1}^{T} \pi(\mathbf{a}_t | \mathbf{x}_{1:t}, \mathbf{a}_{1:t-1}).
$$
(11)

As discussed by Levine [42], the agent does not have control over the stochastic dynamics, so we use the dynamics $p(\mathbf{z}_{t+1} | \mathbf{z}_t, \mathbf{a}_t)$ for $t \geq \tau + 1$ in the variational distribution in order to prevent the agent from choosing optimistic actions.

The joint likelihood is

$$
p(\mathbf{x}_{1:\tau+1}, \mathcal{O}_{\tau+1:T}, \mathbf{z}_{1:T}, \mathbf{a}_{\tau+1:T} | \mathbf{a}_{1:\tau}) = \prod_{t=1}^{\tau+1} p(\mathbf{x}_t | \mathbf{z}_t) \prod_{t=0}^{T-1} p(\mathbf{z}_{t+1} | \mathbf{z}_t, \mathbf{a}_t) \prod_{t=\tau+1}^{T} p(\mathcal{O}_t | \mathbf{z}_t, \mathbf{a}_t) \prod_{t=\tau+1}^{T} p(\mathbf{a}_t).
$$
(12)

We use the posterior from Equation (11), the likelihood from Equation (12), and Jensen's inequality to obtain the ELBO of the marginal likelihood,

$$
\log p(\mathbf{x}_{1:\tau+1}, \mathcal{O}_{\tau+1:T} | \mathbf{a}_{1:\tau})
$$

$$
= \log \int_{\mathbf{z}_{1:T}} \int_{\mathbf{a}_{\tau+1:T}} p(\mathbf{x}_{1:\tau+1}, \mathcal{O}_{\tau+1:T}, \mathbf{z}_{1:T}, \mathbf{a}_{\tau+1:T} | \mathbf{a}_{1:\tau}) \, \mathrm{d}\mathbf{z}_{1:T} \, \mathrm{d}\mathbf{a}_{\tau+1:T}
\tag{13}
$$

$$
\geq \mathop{\mathbb{E}}_{(\mathbf{z}_{1:T}, \mathbf{a}_{\tau+1:T}) \sim q} \left[ \log p(\mathbf{x}_{1:\tau+1}, \mathcal{O}_{\tau+1:T}, \mathbf{z}_{1:T}, \mathbf{a}_{\tau+1:T} | \mathbf{a}_{1:\tau}) - \log q(\mathbf{z}_{1:T}, \mathbf{a}_{\tau+1:T} | \mathbf{x}_{1:\tau+1}, \mathbf{a}_{1:\tau}) \right]
\tag{14}
$$

$$
= \mathop{\mathbb{E}}_{(\mathbf{z}_{1:T}, \mathbf{a}_{\tau+1:T}) \sim q} \left[ \underbrace{\sum_{t=0}^{\tau} \Big( \log p(\mathbf{x}_{t+1} | \mathbf{z}_{t+1}) - \mathrm{D}_{\mathrm{KL}}(q(\mathbf{z}_{t+1} | \mathbf{x}_{t+1}, \mathbf{z}_t, \mathbf{a}_t) \,\|\, p(\mathbf{z}_{t+1} | \mathbf{z}_t, \mathbf{a}_t)) \Big)}_{\text{model objective terms}} \right.
$$
$$
\left. + \underbrace{\sum_{t=\tau+1}^{T} \Big( r(\mathbf{z}_t, \mathbf{a}_t) + \log p(\mathbf{a}_t) - \log \pi(\mathbf{a}_t | \mathbf{x}_{1:t}, \mathbf{a}_{1:t-1}) \Big)}_{\text{policy objective terms}} \right],
\tag{15}
$$

We are interested in the likelihood of optimal trajectories, so we use $\mathcal{O}_t = 1$ for $t \geq \tau + 1$, and its distribution is given by $p(\mathcal{O}_t = 1 | \mathbf{z}_t, \mathbf{a}_t) = \exp(r(\mathbf{z}_t, \mathbf{a}_t))$ in the control as inference framework.

Notice that the dynamics terms $\log p(\mathbf{z}_{t+1}|\mathbf{z}_t, \mathbf{a}_t)$ for $t \geq \tau + 1$ from the posterior and the prior cancel each other out in the ELBO.

The first part of the ELBO corresponds to the model objective. When using the parametric function approximators, the negative of it corresponds directly to the model loss in Equation (7).

The second part of the ELBO corresponds to the maximum entropy RL objective. We assume a uniform action prior, so the $\log p(\mathbf{a}_t)$ term is a constant term that can be omitted when optimizing this objective. We use message passing to optimize this objective, with messages defined as

$$Q(\mathbf{z}_t, \mathbf{a}_t) = r(\mathbf{z}_t, \mathbf{a}_t) + \underset{\mathbf{z}_{t+1} \sim q(\cdot|\mathbf{x}_{t+1}, \mathbf{z}_t, \mathbf{a}_t)}{\mathbb{E}} \Big[ V(\mathbf{z}_{t+1}) \Big] \tag{16}$$

$$V(\mathbf{z}_t) = \log \int_{\mathbf{a}_t} \exp(Q(\mathbf{z}_t, \mathbf{a}_t)) \, \mathrm{d}\mathbf{a}_t. \tag{17}$$

Then, the maximum entropy RL objective can be expressed in terms of the messages as

$$\underset{(\mathbf{z}_{\tau+1:T}, \mathbf{a}_{\tau+1:T}) \sim q}{\mathbb{E}} \left[ \sum_{t=\tau+1}^{T} \Big( r(\mathbf{z}_t, \mathbf{a}_t) - \log \pi(\mathbf{a}_t|\mathbf{x}_{1:t}, \mathbf{a}_{1:t-1}) \Big) \right]$$

$$= \underset{\mathbf{z}_{\tau+1} \sim q(\cdot|\mathbf{x}_{\tau+1}, \mathbf{z}_\tau, \mathbf{a}_\tau)}{\mathbb{E}} \left[ \underset{\mathbf{a}_{\tau+1} \sim \pi(\cdot|\mathbf{x}_{1:\tau+1}, \mathbf{a}_{1:\tau})}{\mathbb{E}} \Big[ Q(\mathbf{z}_{\tau+1}, \mathbf{a}_{\tau+1}) - \log \pi(\mathbf{a}_{\tau+1}|\mathbf{x}_{1:\tau+1}, \mathbf{a}_{1:\tau}) \Big] \right] \tag{18}$$

$$= \underset{\mathbf{a}_{\tau+1} \sim \pi(\cdot|\mathbf{x}_{1:\tau+1}, \mathbf{a}_{1:\tau})}{\mathbb{E}} \left[ \underset{\mathbf{z}_{\tau+1} \sim q(\cdot|\mathbf{x}_{\tau+1}, \mathbf{z}_\tau, \mathbf{a}_\tau)}{\mathbb{E}} \Big[ Q(\mathbf{z}_{\tau+1}, \mathbf{a}_{\tau+1}) \Big] - \log \pi(\mathbf{a}_{\tau+1}|\mathbf{x}_{1:\tau+1}, \mathbf{a}_{1:\tau}) \right] \tag{19}$$

$$= -\mathrm{D}_{\mathrm{KL}} \left( \pi(\mathbf{a}_{\tau+1}|\mathbf{x}_{1:\tau+1}, \mathbf{a}_{1:\tau}) \, \middle\| \, \frac{\exp\left(\mathbb{E}_{\mathbf{z}_{\tau+1}\sim q}\left[Q(\mathbf{z}_{\tau+1}, \mathbf{a}_{\tau+1})\right]\right)}{\exp\left(\mathbb{E}_{\mathbf{z}_{\tau+1}\sim q}\left[V(\mathbf{z}_{\tau+1})\right]\right)} \right) + \underset{\mathbf{z}_{\tau+1}\sim q}{\mathbb{E}} \Big[ V(\mathbf{z}_{\tau+1}) \Big], \tag{20}$$

where the first equality is obtained from dynamic programming (see Levine [42] for details), the second equality is obtain by swapping the order of the expectations, the third from the definition of KL divergence, and $\mathbb{E}_{\mathbf{z}_t \sim q}\left[V(\mathbf{z}_t)\right]$ is the normalization factor for $\mathbb{E}_{\mathbf{z}_t \sim q}\left[Q(\mathbf{z}_t, \mathbf{a}_t)\right]$ with respect to $\mathbf{a}_t$. Since the KL divergence term is minimized when its two arguments represent the same distribution, the optimal policy is given by

$$\pi(\mathbf{a}_t|\mathbf{x}_{1:t}, \mathbf{a}_{1:t-1}) = \exp\left( \underset{\mathbf{z}_t \sim q}{\mathbb{E}} \Big[ Q(\mathbf{z}_t, \mathbf{a}_t) - V(\mathbf{z}_t) \Big] \right). \tag{21}$$

That is, the optimal policy is optimal with respect to the expectation over the belief of the Q value of the learned MDP. This is equivalent to the Q-MDP heuristic, which amounts to assuming that any uncertainty in the belief is gone after the next action [43].

Noting that the KL divergence term is zero for the optimal action, the equality from Equation (18) and Equation (20) can be used in Equation (16) to obtain

$$Q(\mathbf{z}_t, \mathbf{a}_t) = r(\mathbf{z}_t, \mathbf{a}_t) + \underset{\mathbf{z}_{t+1} \sim q(\cdot|\mathbf{x}_{t+1}, \mathbf{z}_t, \mathbf{a}_t)}{\mathbb{E}} \left[ \underset{\mathbf{a}_{t+1} \sim \pi(\cdot|\mathbf{x}_{1:t+1}, \mathbf{a}_{1:t})}{\mathbb{E}} \Big[ Q(\mathbf{z}_{t+1}, \mathbf{a}_{t+1}) \right.$$
$$\left. - \log \pi(\mathbf{a}_{t+1}|\mathbf{x}_{1:t+1}, \mathbf{a}_{1:t}) \Big] \right]. \tag{22}$$

This equation corresponds to the Bellman backup with a soft maximization for the value function.

As mentioned in Section 5, our algorithm conditions the parametric policy in the history of observations and actions, which allows us to directly execute the policy without having to perform inference on the latent state at run time. When using the parametric function approximators, the negative of the maximum entropy RL objective, written as in Equation (18), corresponds to the policy loss in Equation (10). Lastly, the Bellman backup of Equation (22) corresponds to the Bellman residual in Equation (9) when approximated by a regression objective.

We showed that the SLAC objectives can be derived from applying variational inference in the control as inference framework in the POMDP setting. This leads to the joint likelihood of the past

observations and future optimality variables, which we aim to optimize by maximizing the ELBO of the log-likelihood. We decompose the ELBO into the model objective and the maximum entropy RL objective. We express the latter in terms of messages of Q-functions, which in turn are learned by minimizing the Bellman residual. These objectives lead to the model, policy, and critic losses.

## B  Latent Variable Factorization and Network Architectures

In this section, we describe the architecture of our sequential latent variable model. Motivated by the recent success of autoregressive latent variables in VAEs [49, 44], we factorize the latent variable $\mathbf{z}_t$ into two stochastic variables, $\mathbf{z}_t^1$ and $\mathbf{z}_t^2$, as shown in Figure 7. This factorization results in latent distributions that are more expressive, and it allows for some parts of the prior and posterior distributions to be shared. We found this design to provide a good balance between ease of training and expressivity, producing good reconstructions and generations and, crucially, providing good representations for reinforcement learning. Note that the diagram in Figure 7

**Figure 7:** Diagram of our full model. Solid arrows show the generative model, dashed arrows show the inference model. Rewards are not shown for clarity.

represents the *Bayes net* corresponding to our full model. However, since all of the latent variables are stochastic, this visualization also presents the design of the computation graph. Inference over the latent variables is performed using amortized variational inference, with all training done via reparameterization. Hence, the computation graph can be deduced from the diagram by treating all solid arrows as part of the generative model and all dashed arrows as part of approximate posterior.

The generative model consists of the following probability distributions:

$$\mathbf{z}_1^1 \sim p(\mathbf{z}_1^1)$$
$$\mathbf{z}_1^2 \sim p_\psi(\mathbf{z}_1^2|\mathbf{z}_1^1)$$
$$\mathbf{z}_{t+1}^1 \sim p_\psi(\mathbf{z}_{t+1}^1|\mathbf{z}_t^2,\mathbf{a}_t)$$
$$\mathbf{z}_{t+1}^2 \sim p_\psi(\mathbf{z}_{t+1}^2|\mathbf{z}_{t+1}^1,\mathbf{z}_t^2,\mathbf{a}_t)$$
$$\mathbf{x}_t \sim p_\psi(\mathbf{x}_t|\mathbf{z}_t^1,\mathbf{z}_t^2)$$
$$r_t \sim p_\psi(r_t|\mathbf{z}_t^1,\mathbf{z}_t^2,\mathbf{a}_t,\mathbf{z}_{t+1}^1,\mathbf{z}_{t+1}^2).$$

The initial distribution $p(\mathbf{z}_1^1)$ is a multivariate standard normal distribution $\mathcal{N}(\mathbf{0},\boldsymbol{I})$. All of the other distributions are conditional and parameterized by neural networks with parameters $\psi$. The networks for $p_\psi(\mathbf{z}_1^2|\mathbf{z}_1^1)$, $p_\psi(\mathbf{z}_{t+1}^1|\mathbf{z}_t^2,\mathbf{a}_t)$, $p_\psi(\mathbf{z}_{t+1}^2|\mathbf{z}_{t+1}^1,\mathbf{z}_t^2,\mathbf{a}_t)$, and $p_\psi(r_t|\mathbf{z}_t,\mathbf{a}_t,\mathbf{z}_{t+1})$ consist of two fully connected layers, each with 256 hidden units, and a Gaussian output layer. The Gaussian layer is defined such that it outputs a multivariate normal distribution with diagonal variance, where the mean is the output of a linear layer and the diagonal standard deviation is the output of a fully connected layer with softplus non-linearity. The pre-transformed standard deviation right before the softplus non-linearity is gradient clipped element-wise by value to within $[-10,10]$ during the backward pass. The observation model $p_\psi(\mathbf{x}_t|\mathbf{z}_t)$ consists of 5 transposed convolutional layers (256 $4 \times 4$, 128 $3 \times 3$, 64 $3 \times 3$, 32 $3 \times 3$, and 3 $5 \times 5$ filters, respectively, stride 2 each, except for the first layer). The output variance for each image pixel is fixed to a constant $\sigma^2$, which is a hyperparameter $\sigma^2 \in \{0.04, 0.1, 0.4\}$ on DeepMind Control Suite and $\sigma^2 = 0.1$ on OpenAI Gym.

The variational distribution $q$, also referred to as the inference model or the posterior, is represented by the following factorization:

$$\mathbf{z}_1^1 \sim q_\psi(\mathbf{z}_1^1|\mathbf{x}_1)$$
$$\mathbf{z}_1^2 \sim p_\psi(\mathbf{z}_1^2|\mathbf{z}_1^1)$$
$$\mathbf{z}_{t+1}^1 \sim q_\psi(\mathbf{z}_{t+1}^1|\mathbf{x}_{t+1},\mathbf{z}_t^2,\mathbf{a}_t)$$
$$\mathbf{z}_{t+1}^2 \sim p_\psi(\mathbf{z}_{t+1}^2|\mathbf{z}_{t+1}^1,\mathbf{z}_t^2,\mathbf{a}_t).$$

The networks representing the distributions $q_\psi(\mathbf{z}_1^1|\mathbf{x}_1)$ and $q_\psi(\mathbf{z}_{t+1}^1|\mathbf{x}_{t+1},\mathbf{z}_t^2,\mathbf{a}_t)$ both consist of 5 convolutional layers (32 $5 \times 5$, 64 $3 \times 3$, 128 $3 \times 3$, 256 $3 \times 3$, and 256 $4 \times 4$ filters, respectively, stride 2 each, except for the last layer), 2 fully connected layers (256 units each), and a Gaussian output layer. The parameters of the convolution layers are shared among both distributions.

Note that the variational distribution over $\mathbf{z}_1^2$ and $\mathbf{z}_{t+1}^2$ is intentionally chosen to exactly match the generative model $p$, such that this term does not appear in the KL-divergence within the ELBO, and a separate variational distribution is only learned over $\mathbf{z}_1^1$ and $\mathbf{z}_{t+1}^1$. In particular, the KL-divergence over $\mathbf{z}_{t+1}$ simplifies to the KL-divergence over $\mathbf{z}_{t+1}^1$:

$$\mathrm{D}_{\mathrm{KL}}\big(q(\mathbf{z}_{t+1}|\mathbf{x}_{t+1},\mathbf{z}_t,\mathbf{a}_t)\ \|\ p(\mathbf{z}_{t+1}|\mathbf{z}_t,\mathbf{a}_t)\big) \tag{23}$$

$$= \underset{\mathbf{z}_{t+1}\sim q(\cdot|\mathbf{x}_{t+1},\mathbf{z}_t,\mathbf{a}_t)}{\mathbb{E}}\Big[\log q(\mathbf{z}_{t+1}|\mathbf{x}_{t+1},\mathbf{z}_t,\mathbf{a}_t) - \log p(\mathbf{z}_{t+1}|\mathbf{z}_t,\mathbf{a}_t)\Big] \tag{24}$$

$$= \underset{\mathbf{z}_{t+1}^1\sim q(\cdot|\mathbf{x}_{t+1},\mathbf{z}_t^2,\mathbf{a}_t)}{\mathbb{E}}\Bigg[\underset{\mathbf{z}_{t+1}^2\sim p(\cdot|\mathbf{z}_{t+1}^1,\mathbf{z}_t^2,\mathbf{a}_t)}{\mathbb{E}}\Big[\log q(\mathbf{z}_{t+1}^1|\mathbf{x}_{t+1},\mathbf{z}_t^2,\mathbf{a}_t) \tag{25}$$

$$+ \log p(\mathbf{z}_{t+1}^2|\mathbf{z}_{t+1}^1,\mathbf{z}_t^2,\mathbf{a}_t) - \log p(\mathbf{z}_{t+1}^1|\mathbf{z}_t^2,\mathbf{a}_t) - \log p(\mathbf{z}_{t+1}^2|\mathbf{z}_{t+1}^1,\mathbf{z}_t^2,\mathbf{a}_t)\Big]\Bigg]$$

$$= \underset{\mathbf{z}_{t+1}^1\sim q(\cdot|\mathbf{x}_{t+1},\mathbf{z}_t^2,\mathbf{a}_t)}{\mathbb{E}}\Big[\log q(\mathbf{z}_{t+1}^1|\mathbf{x}_{t+1},\mathbf{z}_t^2,\mathbf{a}_t) - \log p(\mathbf{z}_{t+1}^1|\mathbf{z}_t^2,\mathbf{a}_t)\Big] \tag{26}$$

$$= \mathrm{D}_{\mathrm{KL}}\big(\log q(\mathbf{z}_{t+1}^1|\mathbf{x}_{t+1},\mathbf{z}_t^2,\mathbf{a}_t)\ \|\ \log p(\mathbf{z}_{t+1}^1|\mathbf{z}_t^2,\mathbf{a}_t)\big). \tag{27}$$

This intentional design decision simplifies the training process.

The latent variables have 32 and 256 dimensions, respectively, i.e. $\mathbf{z}_t^1 \in \mathbb{R}^{32}$ and $\mathbf{z}_t^2 \in \mathbb{R}^{256}$. For the image observations, $\mathbf{x}_t \in [0,1]^{64\times64\times3}$. All the layers, except for the output layers, use leaky ReLU non-linearities. Note that there are no deterministic recurrent connections in the network—all networks are feedforward, and the temporal dependencies all flow through the stochastic units $\mathbf{z}_t^1$ and $\mathbf{z}_t^2$.

For the reinforcement learning process, we use a critic network $Q_\theta$ consisting of 2 fully connected layers (256 units each) and a linear output layer. The actor network $\pi_\phi$ consists of 5 convolutional layers, 2 fully connected layers (256 units each), a Gaussian layer, and a tanh bijector, which constrains the actions to be in the bounded action space of $[-1,1]$. The convolutional layers are shared with the ones from the latent variable model, but the parameters of these layers are only updated by the model objective and not by the actor objective.

## C   Training and Evaluation Details

Before the agent starts learning on the task, the model is first pretrained using a small amount of random data. The DeepMind Control Suite experiments pretrains the model for 50000 iterations, using random data from 10 episodes, and random actions that are sampled from a tanh-transformed Gaussian distribution with zero mean and a scale of 2, i.e. $\mathbf{a} = \tanh\tilde{\mathbf{a}}$, where $\tilde{\mathbf{a}} \sim \mathcal{N}(0,2^2)$. The OpenAI Gym experiments pretrains the model for 100000 iterations, using random data from 10000 agent steps, and uniformly distributed random actions. Note that this data is taken into account in our plots.

The control portion of our algorithm uses the same hyperparameters as SAC [24], except for a smaller replay buffer size of 100000 environment steps (instead of a million) due to the high memory usage of image observations.

The network parameters are initialized using the default initialization distributions. In the case of the DeepMind Control Suite experiments, the scale of the policy's pre-transformed Gaussian distribution is scaled by 2. This, as well as the initial tanh-transformed Gaussian policy, contributes to trajectories with larger actions (i.e. closer to $-1$ and 1) at the beginning of training. This didn't make a difference for the DeepMind Control Suite tasks *except* for the walker task, where we observed that this initialization resulted in less variance across trials and avoided trials that would otherwise get stuck in local optima early in training.

All of the parameters are trained with the Adam optimizer [37], and we perform 1 gradient step per environment step for DeepMind Control Suite and 3 gradient steps per environment step for OpenAI Gym. The Q-function and policy parameters are trained with a learning rate of 0.0003 and a batch size of 256. The model parameters are trained with a learning rate of 0.0001 and a batch size of 32. We use fixed-length sequences of length 8, rather than all the past observations and actions within the episode.

| Benchmark | Task | Action repeat | Original control time step (s) | Effective control time step (s) |
|---|---|---|---|---|
| DeepMind Control Suite | Cheetah, run | 4 | 0.01 | 0.04 |
| | Walker, walk | 2 | 0.025 | 0.05 |
| | Ball in cup, catch | 4 | 0.02 | 0.08 |
| | Finger, spin | 1 | 0.02 | 0.02 |
| | Cartpole, swingup | 4 | 0.01 | 0.04 |
| | Reacher, easy | 4 | 0.02 | 0.08 |
| OpenAI Gym | HalfCheetah-v2 | 1 | 0.05 | 0.05 |
| | Walker2d-v2 | 4 | 0.008 | 0.032 |
| | Hopper-v2 | 2 | 0.008 | 0.016 |
| | Ant-v2 | 4 | 0.05 | 0.2 |

**Table 1:** Action repeats and the corresponding agent's control time step used in our experiments.

We use action repeats for all the methods, except for D4PG for which we use the reported results from prior work [52]. The number of environment steps reported in our plots correspond to the unmodified steps of the benchmarks. Note that the methods that use action repeats only use a fraction of the environment steps reported in our plots. For example, 1 million environment steps of the cheetah task correspond to 250000 samples when using an action repeat of 4. The action repeats used in our experiments are given in Table 1.

Unlike in prior work [24, 25], we use the same stochastic policy as both the behavioral and evaluation policy since we found the deterministic greedy policy to be comparable or worse than the stochastic policy.

Our plots show results over multiple trials (i.e. seeds), and each trial computes average returns from 10 evaluation episodes. We used 10 trials for the DeepMind Control Suite experiments and 5 trials for the OpenAI Gym experiments. In the case of the DeepMind Control Suite experiments, we sweep over $\sigma^2 \in \{0.04, 0.1, 0.4\}$ and plot the results corresponding to the hyperparameter $\sigma^2$ that achieves the best per-task average return across trials averaged over the first half a million environment steps. In Figure 3, the best $\sigma^2$ values are 0.1, 0.4, 0.04, and 0.1 for the cheetah run, walker walk, ball-in-cup catch, and finger spin tasks, respectively.

## D   Ablation Experiments

We show results for the ablation experiments from Section 6.2 for additional environments. Figure 8 compares different design choices for the latent variable model. Figure 9 compares alternative choices for the actor and critic inputs as either the observation-action history or the latent sample. Figure 10 compares the effect of pretraining the model before the agent starts learning on the task. Figure 11 compares the effect of the number of training updates per iteration. In addition, we investigate the choice of the decoder output variance and using random cropping for data augmentation.

As in the main DeepMind Control Suite results, all the ablation experiments sweep over $\sigma^2 \in \{0.04, 0.1, 0.4\}$ and show results corresponding to the best per-task hyperparameter $\sigma^2$, unless otherwise specified. This ensures a fairer and more informative comparison across the ablated methods.

**Output variance of the pixel decoder.** The output variance $\sigma^2$ of the pixels in the image determines the relative weighting between the reconstruction loss and the KL-divergence. The best weighting is determined by the complexity of the dataset [3], which in our case is dependent on the task.

As shown in Figure 12, our model is sensitive to this hyperparameter, just as with any other VAE model. Overall, a value of $\sigma^2 = 0.1$ gives good results, except for the walker walk and ball-in-cup catch tasks. The walker walk task benefits from a larger $\sigma^2 = 0.4$ likely because the images are harder to predict, a larger pixel area of the image changes over time, and the walker configuration varies considerably within an episode and throughout learning (e.g. when the walker falls over and bounces off the ground). On the other hand, the ball-in-cup catch task benefits from a smaller $\sigma^2 = 0.04$ likely because fewer pixels change over time.

**Random cropping.** We next investigate the effect of using random cropping for data augmentation. This augmentation consists of padding (replication) the $64 \times 64$ images by 4 pixels on each side, resulting in $72 \times 72$ images, and randomly sampling $64 \times 64$ crops from them. For training, we use these randomly translated images both as inputs to the model and the policy, whereas we use the original images as targets for the reconstruction loss of the model. For evaluation, we always use the original images as inputs to the policy.

As shown in Figure 13, this random cropping doesn't improve the learning performance except for the reacher easy task, in which this data augmentation results in faster learning and higher asymptotic performance.

**Figure 8:** Comparison of different design choices for the latent variable model. In all cases, we use the RL framework of SLAC and only vary the choice of model for representation learning. These results show that including temporal dependencies leads to the largest improvement in performance, followed by the autoregressive latent variable factorization and using a fully stochastic model.

**Figure 9:** Comparison of alternative choices for the actor and critic inputs as either the observation-action history or the latent sample. With the exception of the cartpole swingup and reacher easy tasks, the performance is significantly worse when the critic input is the history instead of the latent sample, and indifferent to the choice for the actor input.

**Figure 10:** Comparison of the effect of pretraining the model before the agent starts learning on the task. These results show that the agent benefits from the supervision signal of the model even before making any progress on the task—little or no pretraining results in slower learning and, in some cases, worse asymptotic performance.

**Figure 11:** Comparison of the effect of the number of training updates per iteration (i.e. training updates per environment step). These results show that more training updates per iteration speeds up learning slightly, but too many updates per iteration causes higher variance across trials and, in some cases, slightly worse asymptotic performance.

**Figure 12:** Comparison of different choices for the output variance of the pixel decoder. Good performance is achieved with $\sigma^2 = 0.1$, except for the tasks walker walk ($\sigma^2 = 0.4$) and ball-in-cup catch ($\sigma^2 = 0.04$).

**Figure 13:** Comparison of using random cropping for data augmentation of the input images. The random cropping doesn't improve the learning performance except for the reacher easy task, in which this data augmentation results in faster learning and higher asymptotic performance.

## E   Predictions from the Latent Variable Model

We show example image samples from our learned sequential latent variable model in Figure 14 and Figure 15. Samples from the posterior show the images $\mathbf{x}_t$ as constructed by the decoder $p_\psi(\mathbf{x}_t|\mathbf{z}_t)$, using a sequence of latents $\mathbf{z}_t$ that are encoded and sampled from the posteriors, $q_\psi(\mathbf{z}_1|\mathbf{x}_1)$ and $q_\psi(\mathbf{z}_{t+1}|\mathbf{x}_{t+1}, \mathbf{z}_t, \mathbf{a}_t)$. Samples from the prior, on the other hand, use a sequence of latents where $\mathbf{z}_1$ is sampled from $p_\psi(\mathbf{z}_1)$ and all remaining latents $\mathbf{z}_t$ are from the propagation of the previous latent state through the latent dynamics $p_\psi(\mathbf{z}_{t+1}|\mathbf{z}_t, \mathbf{a}_t)$. Note that these prior samples do not use any image frames as inputs, and thus they do not correspond to any ground truth sequence. We also show samples from the conditional prior, which is conditioned on the first image from the true sequence: for this, the sampling procedure is the same as the prior, except that $\mathbf{z}_1$ is encoded and sampled from the posterior $q_\psi(\mathbf{z}_1|\mathbf{x}_1)$, rather than being sampled from $p_\psi(\mathbf{z}_1)$. We notice that the generated images samples can be sharper and more realistic by using a smaller variance for $p_\psi(\mathbf{x}_t|\mathbf{z}_t)$ when training the model, but at the expense of a representation that leads to lower returns. Finally, note that we do not actually use the samples from the prior for training.

**Figure 14:** Example image sequences for the four DeepMind Control Suite tasks (first rows), along with corresponding posterior samples (reconstruction) from our model (second rows), and generated predictions from the generative model (last two rows). The second to last row is conditioned on the first frame (i.e., the posterior model is used for the first time step while the prior model is used for all subsequent steps), whereas the last row is not conditioned on any ground truth images. Note that all of these sampled sequences are conditioned on the same action sequence, and that our model produces highly realistic samples, even when predicting via the generative model.

**Figure 15:** Example image sequences for the four OpenAI Gym tasks (first rows), along with corresponding posterior samples (reconstruction) from our model (second rows), and generated predictions from the generative model (last two rows). The second to last row is conditioned on the first frame (i.e., the posterior model is used for the first time step while the prior model is used for all subsequent steps), whereas the last row is not conditioned on any ground truth images. Note that all of these sampled sequences are conditioned on the same action sequence, and that our model produces highly realistic samples, even when predicting via the generative model.