[Reviews · NeurIPS 2020]

Review 1

Summary and Contributions: The paper presents a method for estimation of the underlying POMDP of an RL problem for subsequent solution it with a parametrised policy. The model of the POMDP is used to obtain a state estimator/filter which can then be used to train both a critic and an actor with the samples from the filter as inputs, resulting in a model-free algorithm. The method is heavily based on the family of sequential latent variable models trained with variational inference.

Strengths: The paper is solid. The writing is done well and the method theoretically sound. The results are convincing. The ablation studies are tremendously useful for practicioners and have helped me drive design decisions.

Weaknesses: - The paper's narrative is based around POMDPs, but the experimental evaluation does not really stress the capability of the method in that respect. Evaluation is done on pixel-based control, which is PO of course, but we have know that a lagged observation of a few time-steps can make the state fully observable quickly. (See the appendix of [1]). Hence, we do not know how the method fares in environments where the state uncertainty has to be actively reduced by the agent. Therefore I think the paper overstates the results. It is easy to get out of this, however, since one can just drop the POMDP claim. - The justification of the overall approach could have been improved. For me personally (and the optimal control community) it is obvious that we want some kind of state estimation when we use control, as most–if not all–practical problems are PO. But the paper could have done a much better job at its justification. E.g. a very noisy sensor that requires a few time steps waiting to correctly estimate a quantity makes such approaches necessary. The authors suffer from the fact that the RL community is somewhat focused on Mujoco-like benchmarks, which are representative of only a very small fraction of practical optimal control problems. But the authors could have chosen to use a different suite of environments, such as EscapeRoomba or MountainHike, which would illustrate this. If the authors had chosen to conduct experiments that tackle much more relevant POMDP problems, I'd have given an increased score. - I would have enjoyed an ablation whether AISOC/MaxEnt is necessary. [1] **CURL: Contrastive Unsupervised Representations for Reinforcement Learning** Michael Laskin*, Aravind Srinivas*, Pieter Abbeel. Thirty-seventh International Conference Machine Learning (ICML), 2020.

Correctness: The paper claims SOTA at various points. I know this was the case during the first submissions of this paper and at the time of writing, but right now I think one cannot ignore [1, 2, 3]. I feel sorry for the authors because this is just because of publications in the mean-time, but as of now the claims in the paper are wrong. The manuscript has to correct this, as this clearly stands in the way of publication. (I am willing to increase my score radically if this point is adressed–I don't think SOTA results are very relevant for the publication, I think factual correctness is.) [1] **CURL: Contrastive Unsupervised Representations for Reinforcement Learning** Michael Laskin*, Aravind Srinivas*, Pieter Abbeel. Thirty-seventh International Conference Machine Learning (ICML), 2020. [2] **Reinforcement Learning with Augmented Data** Michael Laskin*, Kimin Lee*, Adam Stooke, Lerrel Pinto, Pieter Abbeel, Aravind Srinivas [3] Kostrikov et al. [Image Augmentation Is All You Need: Regularizing Deep Reinforcement Learning from Pixels](https://arxiv.org/abs/2004.13649). arXiv 2020.

Clarity: Yes, very well.

Relation to Prior Work: [4] is missing from the related work. It was one of (if not the) first works using a amortized variational sequence model in a control context. Karl, Maximilian, et al. "Unsupervised real-time control through variational empowerment." arXiv preprint arXiv:1710.05101 (2017).

Reproducibility: Yes

Additional Feedback: It feels as if the work was merely resubmitted. I think it could be improved in the light of recent findings, and would still be relevant. E.g. an experimental evaluation on environments that [1, 2, 3] clearly cannot solve, would greatly increase the relevancy of SLAC.


Review 2

Summary and Contributions: The authors propose learning a (generative) latent variable model alongside the model-free q based approach. The authors show that this lead to improved performance on standard robotic bechmarks

Strengths: - The writing is very good including the theoretical work and description of experiments - the experiments and especially the ablation studies make sense given the research question

Weaknesses: - A comparison to E2C would have been nice - See my general comments on additional feedbacks about separating the policy learning from the embedding / state transition learning. - The results of the proposed method appear a bit unstable e.g for Walker or Hopper. - By the results from cheetah or Ant v2 I get a bit the impression that the results can be divergent and "early stopping" may be necessary. Why this may be?

Correctness: yes

Clarity: yes

Relation to Prior Work: The authors did a great job covering related work

Reproducibility: Yes

Additional Feedback: I have some doubts over the idea to separate the RL task from the 'learning an embedding and model' part. Isn't one of the main advantages of model-free RL over model-based RL the -- for a lack of better word -- task efficency or attention mechnanism in the sense that only those parts of the system behavior is learned that is useful in saving the task? (compared to model-based RL that tries to learn the whole dynamics). As it is right now the latent variable model (the embedding + state transition model) does not receive any feedback about how its learning benefits the general task. Isn't the "correct" latent representation task dependent?


Review 3

Summary and Contributions: This paper proposes an approach for solving POMDP with rich / high dimensional observations, e.g., images. This approach frames the policy optimization problem as an inference problem whose goal is to maximize the joint likelihood of optimal policy and observations, over an explicit latent state model and a policy model. This directly draws inspiration from structure variational inference and RL as probabilistic inference. The approach is instantiated and experimented on continuous control tasks. And the results show that the new method outperforms some popular model-free and model-based RL algorithms.

Strengths: The new method achieves convincingly good results. The experimental results are comprehensive. Besides showing that the new method achieves superior or similar performance compared to some popular RL algorithms, the authors conducted ablation analysis, which is very helpful to understand the effect of the design choices.The paper contains some useful lessons learned, which provides interesting insights for making design choices in the future, e.g., Line 228, Line 320.

Weaknesses: Algorithmic contribution: The contribution in terms of the algorithm is limited. The two main components of the algorithm can be drawn from existing methods in a straightforward way: the Actor critic part (Line 201) seems to be the same as Levine [40], except that state becomes the latent state and state dependent policy becomes observation dependent policy. And the latent variable model part (Line 190) has been proposed before. Overly optimistic / risk seeking: As mentioned in Levine [40], treating both policy and transition model as variational parameters can lead to risk-seeking behavior. The authors argue that having policies that are dependent on the history of observations instead of the latent state can mitigate this issue. I am not fully convinced that this is the case, because a latent dynamics model and a latent representation is learned anyway and furthermore the goal is to maximizing p(x, O | a). In addition, the experimental results (Figure (b)) seem not to support the authors' claim.

Correctness: I think empirical methodology is correct. However I have some questions with respect to the method. 1. I am a bit confused by Algorithm 1. What is the difference between t and \tau? What do \theta_1 and \theta_2 represent? 2. In Equation (9), On the left, V_\theta(z_{\tau+1}) is a function of z_{\tau+1}. However the right hand side also depends on x_{1:{\tau+1}}. 3. The likelihood in Line 178 is for a single \tau. Are the likelihood for different \tau summed up for optimization?

Clarity: Yes, overall. However, the layout seems to be overly tightened. More loose layout can make reading easier. Furthermore, I think some potentially useful information is missing, e.g., In Eq.(5) is p(z_{t+1} | z_t, a_t) the true unknown transition probability or a model? During pre-training, can the true state be accessed? is that the "supervision signal"?

Relation to Prior Work: Yes, except that the difference between the new method and Hafner et al. [26] in Line 87 is not well addressed.

Reproducibility: Yes

Additional Feedback: ================ == After Rebuttal == ================ I appreciate the authors' thoughtful response and additional experimental results. However, I am still not fully convinced that the proposed method will not suffer from risk seeking since in each iteration the model is updated (Eq. 7) and is used to forecast to update policy (Eq. 10), if I am not mistaken. Furthermore, as pointed by R1 and acknowledged by the authors, the POMDP narrative lacks support. But I think this paper does have interesting metrics. Good luck with the paper.


Review 4

Summary and Contributions: The authors propose a new actor-critic method where the critic operates on the learned latent space but the policy still operates on the state space by taking in past history (though the policy is trained using this "latent space critic"). This latent representation is learned by training a fully stochastic sequential latent model with VAE loss for maximizing observation likelihoods. The method shows very good performance on standard benchmark pixel-based Mujoco environments.

Strengths: Experimental results shown in this paper (Figure 4 Halfcheetah, Walker2d, Hopper, Ant) are quite strong, compared to other recent methods on this domain such as PlaNet. This paper also includes a neat derivation on the proposed SLAC objective using control as inference framework for POMDP settings.

Weaknesses: Since I don't have expertise in this area and particular task domain, I cannot comment on the novelty of the method.

Correctness: Yes. The empirical methodology is correct. I appreciate the ablations studies on key design choices (on latent vs history conditioning for actor and critic) from Figure 9 in Appendix. It shows that using history (instead of latent vector) also performs competitively as latent actor with the added benefit of fast test time policy deployment.

Clarity: Yes

Relation to Prior Work: Yes

Reproducibility: Yes

Additional Feedback: ----Post rebuttal--- I've read the rebuttals. Thank you for the added clarifications and the position of this paper. (e.g. not claiming SOTA as the main contribution, true state of the environment is never available.) The empirical results shown in the paper are still quite strong

[Author Response · NeurIPS 2020]

We thank the reviewers and are happy they found the paper is well-written (R1, R2, R3, R4), the method is theoretically
sound (R1, R2), the experiments are appropriate and comprehensive (R2, R3, R4), the results are convincing (R1, R3,
R4), and the ablation studies are "tremendously useful" and helpful for making design choices (R1, R2, R3, R4). We
address individual questions below.

**R1: POMDP claims.** We'll update the paper to stress that our method is not equipped to solve the POMDP, e.g.
reducing uncertainty. It was not our intent to claim that it was. Although we use the POMDP formalism to derive our
algorithm, we agree with R1 that our experiments do not address the capability of our method to solve POMDPs.

**R1: SOTA claims.** We'll remove the SOTA claims in light of the recent works CURL, RAD, and DrQ [1]. We strongly
agree that this is distracting and less important than the content of the paper, and we had not intended to present
that as the main contribution. Regarding the results: to our knowledge, DrQ (which came out on 28 April 2020, and
would be reasonably considered concurrent with NeurIPS submissions) is the best among these methods, and SLAC
is comparable to DrQ in the 4 DM control tasks, as shown in Figure 1. Note that the DrQ and CURL papers report
outdated and lower-performing results for our method, compared to the results from this submission. That said, we will
remove claims about state of the art results, and add results for all methods.

**Figure 1:** SLAC (ours) achieves comparable performance as DrQ (Kostrikov et al., 2020 [1]) in the 4 DM control tasks. Note that the
plots include the initial exploration steps (10k for SLAC and 1k for DrQ), in which data is collected from a uniformly random policy.

**R2: Divergence and instability.** We hypothesize that it's caused in part by the constantly changing latent space of the
model and the model overfitting to the most recent data (the replay buffer contains the 100 most recent episodes). We
believe that our method could benefit from early stopping or a schedule for model updates (e.g. having a "target" model,
similar to target networks in RL). This merits further investigation, which we leave for future work.

**R2: Task-dependent representations.** The latent representation does receive some task-dependent information (in
the form of reward prediction), although we acknowledge it could further incorporate more feedback from the value
functions and policy. We chose to keep them separate but we recognize that it's worth investigating futher.

**R3: "The two main components of the algorithm can be drawn from existing methods in a straightforward**
**way."** We acknowledge that the individual components are based on previous work, and we will discuss this more
clearly in the paper. Although the resulting algorithm is simple, we believe that formally establishing a link between
them is important to help us understand the implications of certain design choices. E.g., one interesting finding is that,
as indicated by Equation (21) in the Appendix, the optimal policy of the SLAC objective is optimal with respect to
the expectation over the belief of the Q value of the learned MDP. This is equivalent to the Q-MDP heuristic, which
"amounts to assuming that any uncertainty in the agent's current belief state will be gone after the next action" (Littman
et al., 1995 [2]). We'll update the paper to emphasize these observations.

**R3: "Overly optimistic / risk seeking."** We believe we don't have that same issue since the agent isn't allowed to
control the dynamics of the *future* (i.e., the policy objective in Equation (10) updates only the policy parameters).
Furthermore, the history-dependent policies mitigate a similar issue, except that it's caused by something different:
When executing a latent-conditioned policy, the latent distribution is available but the specific latent state is unknown.
Any attempt to choose any latent (e.g. the mode or a random sample) results in an overly confident agent since the
policy assumes that the given latent is correct. By conditioning the policy in the observations, we instead obtain the
Q-MDP heuristic (see paragraph above). We acknowledge that our empirical evaluation doesn't exemplify this. This
issue doesn't arise if the belief is narrow. We'll update the paper to clarify this.

**R3: Correctness and clarity.** We'll update the paper with clarifications on the points brought up by R3. Most
importantly, the true state of the environment is never available.

**R1: Reference in the related work**: We'll add the missing reference to the paper.

[1] Kostrikov et al. Image Augmentation Is All You Need: Regularizing Deep Reinforcement Learning from Pixels. arXiv:2004.13649, 2020.
[2] Littman et al. Learning policies for partially observable environments: Scaling up. Machine Learning Proceedings, 1995.


[Meta-Review · NeurIPS 2020]

The method targets a model-based approach to solve POMDPs with high-dimensional observation spaces. This problem is tackle by learning jointly about the dynamics of the POMDP and the optimal policy by maximum likelihood using an “RL as inference” type objective. In more detail, the latent space transitions are predicted by an inference model that is trained to maximise an evidence lower bound. The reviewers are mostly positive about the paper. They mention the theoretical soundness of the approach and the quality of writing as well as the empirical set-up and usefulness of the ablations. Some criticisms were also mentioned. For example, is the method applicable to ‘strongly partial observable’ environments? One reviewer suggested the justification of the approach could be made more clear in the paper, and a ablation could be added (about the necessity of AISOC / MaxEnt). Another reviewer questioned the novelty of the training. Finally, R1 strongly felt that the SOTA claim was not (no longer) substantiated. The authors rebuttal addressed many of the points of criticism. They agree that the method is not designed to solve strongly partial observable problem and agree to remove the SOTA claim. Most of the reviewers found that the strengths of the paper outweighed its limitations. The author’s rebuttal has furthermore addressed the points by the more critical reviewer in more detailed. In personal communication, this reviewer has confirmed they are happy with accepting the paper now, but they’ve been unable to change the review in CMT. Thus, I’m happy to recommend the paper for acceptance. I’d ask the authors to update the final version of the paper in the light of the reviews. In particular, I’d like the authors to include the points on the “POMDP claims” and “SOTA claims” discussed in the author response, as well as clarify the “Overly optimistic / risk seeking” aspect. Note: it’s not necessary for the authors to contact the AC with inquiries that summarise the reviews so far.